# Development of Polyclonal Antibodies and a Serological-Based Reverse-Transcription Loop-Mediated Isothermal Amplification (S-RT-LAMP) Assay for Rice Black-Streaked Dwarf Virus Detection in Both Rice and Small Brown Planthopper

**DOI:** 10.3390/v15102127

**Published:** 2023-10-20

**Authors:** Yanhong Hua, Chenwei Feng, Tianxiao Gu, Haoyu Chen, Duxuan Liu, Kai Xu, Kun Zhang

**Affiliations:** 1Department of Plant Protection, College of Plant Protection, Yangzhou University, Yangzhou 225009, China; mx120220848@stu.yzu.edu.cn (Y.H.); mz120221464@stu.yzu.edu.cn (C.F.); mz120211383@stu.yzu.edu.cn (T.G.); mx120230827@stu.yzu.edu.cn (H.C.); mz120231451@stu.yzu.edu.cn (D.L.); 2Joint International Research Laboratory of Agriculture and Agri-Product Safety of Ministry of Education of China, Yangzhou University, Yangzhou 225009, China; 3Jiangsu Key Laboratory for Microbes and Functional Genomics, Jiangsu Engineering and Technology Research Center for Microbiology, College of Life Sciences, Nanjing Normal University, Nanjing 210023, China; xukai@njnu.edu.cn

**Keywords:** rice black-streaked dwarf virus, outer coat protein, polyclonal antibodies, serological detections, serological-based reverse-transcription loop-mediated isothermal amplification

## Abstract

Rice black-streaked dwarf virus (RBSDV) infects rice and maize, and seriously affects rice yields in main rice-producing areas. It can be transmitted via small brown planthopper (SBPH: *Laodelphax striatellus* Fallén). To more rapidly, sensitively, and highly throughput diagnose RBSDV in the wild condition, we first purified the recombinant His-CP^RBSDV^ protein, and prepared the polyclonal antibodies against the His-CP^RBSDV^ protein (PAb-CP^RBSDV^). Based on the PAb-CP^RBSDV^, we developed a series of serological detections, such as Western blot, an enzyme-linked immunosorbent assay (ELISA), and a dot immunoblotting assay (DIBA). Furthermore, we developed a serological-based reverse-transcription loop-mediated isothermal amplification assay (S-RT-LAMP) that could accurately detect RBSDV in the wild. Briefly, the viral genomic dsRNA together with viral CP were precipitated by co-immunoprecipitation using the PAb-CP^RBSDV^, then the binding RNAs were crudely isolated and used for RT-LAMP diagnosis. Using the prepared PAb-CP^RBSDV^, four serology-based detection methods were established to specifically detect RBSDV-infected rice plants or SBPHs in the wild. The method of S-RT-LAMP has also been developed to specifically, high-throughput, and likely detect RBSDV in rice seedlings and SBPHs simultaneously. The antiserum prepared here laid the foundation for the rapid and efficient detection of RBSDV-infected field samples, which will benefit for determination of the virulence rate of the transmission vector SBPH and outbreak and epidemic prediction of RBSDV in a rice production area.

## 1. Introduction

Rice black-streaked dwarf virus (RBSDV) is a plant virus characterized by its double-stranded RNA genome. It belongs to the family *Reoviridae* and falls under the genus *Fijivirus*. RBSDV is primarily known for its ability to infect various plants within the *Gramineae* family, such as rice, maize, barley, and wheat. This viral infection poses a considerable threat to agricultural crops in East Asia, resulting in substantial yield losses [1,2]. RBSDV is primarily spread by a specific insect called the small brown planthopper (SBPH, *Laodelphax striatellus* (Fallén). This transmission occurs on rice plants and is persistent and circulative in the wild. The virus is transmitted when the SBPH feeds on the phloem tissues of both infected and healthy rice plants. However, it should be noted that transmission of RBSDV does not occur through the eggs of the planthopper [3,4]. RBSDV-infected plants exhibit characteristic symptoms including severely dwarfing with leaves that are notably dark-green. Additionally, there maybe a distinct appearance of white waxy or black-streaked swelling, predominantly observed along the veins on the undersides of the leaf blades and sheaths. Furthermore, these infected plants also experience a significant decrease in seed fertility, displaying an alarming decline in reproductive capabilities [5,6]. Since the 1990s, an outbreak of the disease occurred in Jiangsu, Zhejiang and other Provinces in China, especially in 2010, when the disease rate in severely affected areas exceeded 50%, resulting in a sharp decline in the yield of rice [7]. Once infected by RBSDV, there is almost no therapeutic method application for farmers. Additionally, RBSDV can also cause Maize rough dwarf disease and Green dwarf disease of wheat and other viral diseases [8]. Hence, this requires that the producer could detect the virus at the early stage, and take action quickly.

RBSDV genome is composed of ten linear double-stranded RNAs, named according to their migration speed on gel electrophoresis, from slow to fast, S1 to S10, respectively [9]. The genome-wide sequence of the virus consists of 29,141 bp nucleotides, which contains thirteen open reading frames (ORFs), and encodes a total of thirteen proteins, including six structural proteins, such as P1, P2, P3, P4, P8 and P10 [10]. One particular protein of significant importance is the RBSDV P10 protein, which serves as the principal outer capsid protein [11]. This protein is frequently employed as a marker for RBSDV infection during disease diagnosis due to its relatively higher levels of expression in infected tissue [12].

In order to identify and detect RBSDV, several methods have been established previously. These techniques include reverse transcription polymerase chain reaction (RT-PCR) [10], Northern blotting [13], indirect enzyme-linked immunosorbent assay (ID-ELISA) [14], quantitative real time polymerase chain reaction (qRT-PCR) [15], RT-loop-mediated isothermal DNA amplification (RT-LAMP) [16] and single small brown planthopper RT-PCR [14,17]. However, these detection methods require professional technicians to operate and are relatively high cost. In addition, there are few reports on methods for simultaneous large-scale detection of RBSDV in rice seedlings and SBPH in the field, such as the monoclonal antibody-based antigen-coated-plate enzyme-linked immunosorbent assay (ACP-ELISA) and dot enzyme-linked immunosorbent assay (dot-ELISA) [12]. However, other polyclonal antibody-based serological methods and other more accurate assays for large-scale detection of both hosts (rice and SBPHs) in the wild are still lacking. Therefore, there is an urgent need for an RBSDV-specific, sensitive, high-throughput and easy to operate method to detect and manage RBSDV in the wild.

In our research, we utilized the prokaryotic expressed His-CP^RBSDV^ as an antigen to immunize rabbits and successfully obtained potent polyclonal antibody titers against the His-CP^RBSDV^ protein (PAb-CP^RBSDV^). Using the PAb-CP^RBSDV^, a variety of serology-based detection methods for RBSDV-infected rice seedlings and SBPHs were established. Furthermore, the present research suggests that the utilization of a blend of both RT-PCR and Western blot methods holds potential for the accurate identification of RBSDV in rice seedlings and SBPHs. Furthermore, this work presents a new method named S-RT-LAMP which combines PAb-CP^RBSDV^-based immunoprecipitation with RT-LAMP. Based on S-RT-LAMP, we can directly and exactly detect RBSDV-infected rice seedlings and SBPHs simultaneously at a large-scale. Our study provided more RBSDV positive detections as it is based on the specific antibody followed by nucleotides amplification, which will detect faster than traditional detection and is beneficial for taking controlling-actions by farmers in an early stage of RBSDV infection.

## 2. Materials and Methods

### 2.1. Construction of the Vector for Recombinant CP^RBSDV^ Protein Expression

The construction method of the prokaryotic expression vector was as follows, referring to the previous method [18]. Total RNA was extracted from RBSDV-infected rice seedlings and cDNA was generated using Takara PrimeScript RT Reagent Kit reverse transcription (Takara, Dalian, China). The RBSDV coat protein (CP) gene with 1677 bp was amplified by RT-PCR with a pair of primers (Table 1). The PCR amplification also introduced *EcoR* I and *Xho* I restriction endonuclease cleavage sites into the fragment. The obtained fragments and pET28 (a)+ vectors are simultaneously double digested with the corresponding restriction enzymes and then electrochemically purified using the 1% agarose and agarose gel DNA Purification Kit (Takara, Dalian, China) via electrochemical methods. The purified fragment was then cloned into a linearized plasmid, resulting in the final plasmid pET28(a)-CP^RBSDV^. To verify the accuracy of the plasmid sequence and the correct reading frame, Sanger sequencing was performed. DNAman software (Lynnon BioSoft, Version 8) was used to predict the expected size of recombinant CP^RBSDV^ proteins from cloned sequences.

### 2.2. Prokaryotic Expression and Purification of the Recombinant His-CP^RBSDV^ Protein

The correct pET28(a)-CP^RBSDV^ plasmid was introduced into the *E. coli* BL21(DE3) strain. Two positive transformants were selected. Then, we performed the small-scale expression experiment under isopropyl-β-D-thiogalactoside (IPTG) (Sangon Biotech, Shanghai, China) induction. We determined the conditions of the highest expression level in these transformants.

The most promising transformant was selected for the subsequent large-scale protein expression experiment according to the previous method, but was slightly modified in some places [19]. The difference was that 2 mM PMSF at a ratio of 1:100 (*v*/*v*) was added prior to disruption to protect the protein from being destroyed by proteases. During bacterial cells disruption, the conditions of the ultrasonic crusher were changed to 300 W, 20 min total, 90 times per cycle, 2 s working, 1 s interval. Finally, different concentrations of imidazole solution were utilized to elute recombinant His-CP^RBSDV^ into packed nickel columns.

### 2.3. Production of the Polyclonal Antibody against the His-CP^RBSDV^ Protein

The method of obtaining polyclonal antibodies against recombinant His-CP^RBSDV^ proteins (PAb-CP^RBSDV^), referring to the previously described method, was used to obtain PAb-CP^RSV^ [18]. To immunize two New Zealand white rabbits, high concentrations of recombinant His-CP^RBSDV^ were employed as the antigen. Every 7 days, 2 mg of recombinant His-CP^RBSDV^ emulsified with Freund’s complete adjuvant was divided into 4 injections into each rabbit. Blood from the veins of rabbit ear was obtained to assess antibody titers after 30 days-post-immunization (DPI). Finally, crude antiserum was obtained by centrifugation at intervals of 15 DPI. The crude polyclonal antibodies were precipitated and purified by salt fractionation (sodium sulfate, 20%). After dialysis and a series of purification procedures, a high purity PAb-CP^RBSDV^ could be obtained.

### 2.4. RT-PCR Detection of Two Rice Viruses

The rice seedlings and SBPHs that may be infected with RBSDV or RSV were collected from various rice fields located in the outskirts of Yangzhou City, Jiangsu Province. Following the instructions provided by the manufacturer, total RNA of RBSDV/RSV-infected rice seedlings and SBPH was extracted by TRIzol Reagent (Invitrogen, Carlsbad, CA, USA). Briefly, about 0.1 g tissue powder of plants or one head of SBPH were mixed with 1.0 mL or 300 μL TRIzol, respectively, and incubated for 5 min. Finally, the RNA was resuspended in DEPC-treated water, with a volume of 20 μL for plant tissue RNA and 10 μL for insect RNA. The concentration of RNA was measured using a NanoPhotometer N60 (Implen, Munich, Germany). To verify the integrity of the extracted total RNAs, electrophoresis was conducted on a 1.5% agarose gel. The cDNA for isolated RNA was generated using Takara PrimeScript RT Reagent Kit reverse transcription and pairs of primers were employed (Table 1). Based on the previously reported sequence (GenBank accession No. JN191602.1 and JQ927422.1), pairs of primers (Table 1) were designed for specific detection of RBSDV and rice stripe virus (RSV).

### 2.5. Three Detection Methods of the RBSDV with Prepared PAb-CP^RBSDV^

PAb-CP^RBSDV^ was utilized for the serological diagnosis of RBSDV-infected rice seedlings and SBPHs. On the basis of the PAb-CP^RBSDV^, the detection methods of Western blot, dot-blot and ELISA were established by following their conventional protocols with slight adjustments [18,19,20].

For the Western blot assay, fresh rice leaves (0.1 g) and single SBPH were ground in liquid nitrogen with a grinder (Shanghai Jingxin Industrial Development Co., Shanghai, China) and then dissolved in the protein extraction buffer as previously described [18]. At the time of addition of the first antibody, the prepared antibodies were diluted to different concentrations. Then, 0.02% horseradish peroxidase (HRP)-coupled anti-rabbit immunoglobulin (HRP-A) (Sangon Biotech, Shanghai, China) was utilized as a secondary antibody. Finally, a DAB Horseradish Peroxidase Color Development Kit (Beyotime Biotech, Shanghai, China) was utilized for color development, which refer to the developer’s instructions, and then the NC membrane was imaged with the Bio-Rad ChemiDoc™ Touch Imaging System (Bio-Rad, Hercules, CA, USA).

For dot-blot detection, the procedure of protein extraction was similar to the previous description with minor modifications [21]. The purified recombinant protein, known as His-CP^RBSDV^, served as the positive control, while the total protein extracted from healthy rice seedlings was considered the negative control. In brief, 3 μL of sample supernatants were loaded to each square on NC membrane. The rest of the procedure was the same as Western blot. In the first antibody incubation step, PAb-CP^RBSDV^ was diluted to 10^4^-fold.

For ELISA detection, the process of protein extraction from rice seedlings and SBPH was the same as in dot-blot, but there was a slight change in dosage [18]. The supernatant of rice samples (150 μL) and planthopper samples (20 μL) were loaded into 96-well polystyrene plates (Costar, Corning, NY, USA), respectively, and incubated at 37 °C for 2 h. Each sample was replicated three times and averaged. The method of chromogenic reaction and termination reaction were consistent with the method as previously described [19]. The color change in the well was then noticed after the absorbance values were read by the ELISA reader PowerWave XS2 (BioTek Instruments, Santa Clara, CA, USA) at OD_405_ and OD_450_, respectively. In the first antibody incubation step, PAb-CP^RBSDV^ was diluted to different concentrations. In the ELISA assay for the specific detection of RBSDV or RSV-infected plants, both PAb-CP^RBSDV^ diluted to 10^4^-fold, and PAb-CP^RSV^ diluted to 10^5^-fold were added simultaneously.

### 2.6. S-RT-LAMP Detection of the RBSDV with Prepared PAb-CP^RBSDV^

The S-RT-LAMP assay was built and modified from the previous RT-LAMP [16], which took advantage of the serological methods to crudely isolate RNA bounded to the viral CP by immunoprecipitation using the specific PAb-CP^RBSDV^, and to accurately and rapidly detect the presence of RBSDV both in rice and SBPH.

For the S-RT-LAMP assay, 3 g rice samples and a single SBPH were taken and ground with liquid nitrogen, followed by adding extraction buffer (20 mM Tris-HCl, pH 7.5, 1 mM EDTA, 150 mM NaCl, 10% Glycerol, 0.2% NP40, 2% PVP40, 10 mM DTT, 1 × cocktail) at a ratio of 1:3 (*w*/*v*). The extracts were then placed on ice for 40 min after vigorous shaking, during which time the mixture was inverted several times. After centrifugation at 1500× *g* at 4 °C for 20 min, the obtained supernatant was incubated 1 h with protein A/G magnetic beads bound to the prepared PAb-CP^RBSDV^. Subsequently, the beads were collected and washed four times with a washing buffer (20 mM Tris-HCl, pH 7.5, 1 mM EDTA, 150 mM NaCl, 10% Glycerol, 0.2% NP40), and the total bounded RNAs from RBSDV-infected rice and SBPH were then extracted by traditional phenol chloroforms.

We targeted the highly conserved regions of RBSDV genomic sequences, and designed four sets of oligonucleotide primers for the RT-LAMP assay. Primers were designed using PrimerExplorer V5 software (available at http://primerexplorer.jp/lampv5e/index.html, accessed on 15 July 2023) with default settings. The primers (Table 2) were targeted to the S10 genomic, and other details are listed in this table.

For the RT-LAMP assay, the *Bst* II Pro DNA Polymerase Large Fragment (8 U/μL, Vazyme, Winooski, China) Kit was used, following the manufacturer’s method. In brief, 1.1 μg of rice RNAs or 0.2 μg of a single SBPH RNAs were served as a template in the 25 μL reaction mixture. At the same time, 0.5 μL of M-MLV reverse transcriptase (200 U/μL, Fermentas, Burlington, ON, USA) and 0.5 μL of the RNase Inhibitor (40 U/μL, Fermentas, Burlington, ON, USA) were also added to the reaction system. The reaction mixture was incubated at 63 °C for 60 min and then for 5 min at 80 °C. Then, 3 μL of amplification products were routinely analyzed for purity and size using 2% agarose gel electrophoresis and TAE staining. Finally, 2 μL SYBR Green I (Solarbio, Beijing, China) was added to RT-LAMP amplification products with a ratio of 1:1000 (*v*/*v*); the color change and fluorescence were directly observed visually or under a UV lamp (JS-780, Shanghai Peiqing Technology, Shanghai, China) to analyze and evaluate the amplification product.

## 3. Results

### 3.1. Expression and Purification of the Recombinant His-CP^RBSDV^ Protein

The *CP^RBSDV^* coding sequences were cloned from RBSDV-infected rice samples collected in Yangzhou City, Jiangsu Province (Figure 1A), and compared with other *CP^RBSDV^* sequences deposited in NCBI, which generated a phylogenetic tree based on *CP^RBSDV^* coding sequences. The results showed that the coding sequences of the *CP^RBSDV^* were highly conserved worldwide (Appendix A). Therefore, it is suitable to choose the CP^RBSDV^ protein as the target for detection of the RBSDV with different origins.

A small-scale expression test showed that a specific band of approximately 63.7 kDa, consistent with the expected size, appeared on the NC membrane after the Western blot assay, which indicated that the target recombinant protein His-CP^RBSDV^ was expressed (Figure 1B). Due to the relative higher expression level of the target protein generated under IPTG induction by clone 1^#^, comparatively to clone 2^#^, clone 1^#^ was selected for subsequent large-scale expression and purification of the target protein.

In large-scale expression analysis, the His-CP^RBSDV^ bounded on resin can be eluted by step-wise imidazole solution from 60 mM to 400 mM. The highest elution efficiency is at 200 mM and 300 mM, and the contaminants are almost negligible (Figure 1C). In summary, the purified target His-CP^RBSDV^ protein has a relatively high purity and is suitable with the demand for further purification.

### 3.2. Antiserum Production Using the Recombinant His-CP^RBSDV^ Protein

The purified His-CP^RBSDV^ obtained from the affinity column, and subsequently dialyzed, was examined using SDS-PAGE. The analysis revealed that the target protein exhibited both a notably high concentration and exceptional purity (Figure 2A). The titer of the prepared antiserum against His-CP^RBSDV^ was determined by ELISA; the results revealed that it had a high titer and that the OD_450_ reading remained above 0.6 even when diluted 4000-fold (Figure 2B, red line). Further, the target antiserum was diluted according to ratios 1:10^−1^, 1:10^−2^, 1:10^−3^, 1:10^−4^, 1:10^−5^, and 1:10^−6^ for immune-dot blotting, and showed that a clear color reaction was still observed even when it was diluted 10^−5^-fold (Figure 2C). Furthermore, we diluted the His-CP^RBSDV^ with a stepwise increased rate (10^3^, 5 × 10^3^, 10^4^, 2 × 10^4^, and 4 × 10^4^), and used the Western blot analyses to see the specificity of the prepared target antiserum. The results showed that a specific band remained visible with the expected size, even when the antigen (His-CP^RBSDV^) was diluted 4 × 10^4^-fold using the 1 × 10^4^-fold diluted antiserum (Figure 2D). These results demonstrated that the prepared target antiserum (against His-CP^RBSDV^) had a high titer and potential in detecting the RBSDV-infected samples. In summary, ELISA, dot-blot, and Western blot showed that the antiserum against His-CP^RBSDV^ that we prepared was of a high quality and can be applicated in the immunoassay detection of RBSDV. 

### 3.3. RT-PCR, Western Blot and Dot Blot Detection of RBSDV Infection in Rice Plants or SBPHs Using the Prepared PAb-CP^RBSDV^

To establish more sensitive, specific, and user-friendly serological detections for RBSDV, we further purified the polyclonal antibodies against the target His-CP^RBSDV^ (PAb-CP^RBSDV^) in serum by salt fractionation. The majority of the IgG-type antibodies existing in the serum was precipitated. Then, we obtained the PAb-CP^RBSDV^ with high purity and concentration after dissolving the precipitate in normal saline, and then dialyzing with salt overnight.

To test the quality of the prepared PAb-CP^RBSDV^, four rice seedlings that were RBSDV-infected and one healthy rice seedling were collected for the RT-PCR assay with CP^RBSDV^ gene specific pairs of primers (RBSDV/P10/F and RBSDV/P10/R) (Table 1). RT-PCR results showed that the samples 1^#^, 2^#^, 3^#^, and 4^#^ we selected were indeed infected by RBSDV (Figure 3A, upper panel). Then, using the prepared PAb-CP^RBSDV^ for Western blot detection, the results confirmed that these four RBSDV-positive rice seedlings were indeed infected with RBSDV, which indicated the prepared PAb-CP^RBSDV^ can be used in RBSDV detection (Figure 3A, middle panel). Furthermore, we extracted the total proteins of these rice seedlings and subjected them to performance with the dot-blot assay, and the purified recombinant His-CP^RSBSDV^ protein and total protein from the healthy rice seedling were treated as the positive and negative control, respectively (Figure 3A, bottom panel). The results showed that even when PAb-CP^RBSDV^ was diluted 10^4^-fold, we still observed strong color reactions on the NC membrane loaded with the total proteins of the four RBSDV-infected rice seedlings, as well as the positive control, which also indicated that the PAb-CP^RBSDV^ can be used in detection.

In order to determine the specificity of PAb-CP^RBSDV^, we firstly perform RT-PCR to screen for RBSDV or RSV-infected samples from rice producing areas. Fortunately, six rice seedlings that exhibited viral symptoms were indeed infected by the RSV (1^#^, 2^#^, and 3^#^) and RBSDV (4^#^, 5^#^, and 6^#^), respectively (Figure 3B, left, row 1). Then, using these three RSV-positive rice seedlings 1^#^, 2^#^, and 3^#^ and three RBSDV-positive rice seedlings 4^#^, 5^#^, and 6^#^ as materials, the specificity of PAb-CP^RSV^ and PAb-CP^RBSDV^ were further validated by Western blot when both were diluted for 1 × 10^4^-fold. The detection results were consistent with the results of RT-PCR, and PAb-CP^RSV^ was not able to detect the RBSDV, and vice versa (Figure 3B, left, row 2 and row 3). All these results demonstrated that the prepared PAb-CP^RBSDV^ had high specificity and sensitivity for the detection of RBSDV in rice plants.

As we know, RSV and RBSDV were transmitted by SBPH in wild field. To better understand the transmission dynamics of these two viruses, the detection methods of RBSDV and RSV should be established on one single SBPH. Hence, six heads of SBPHs that carried RBSDV or RSV were collected. Firstly, the RT-PCR confirmed that the SBPH 1^#^, 2^#^, and 3^#^ were infected with RSV, and the 4^#^, 5^#^, and 6^#^ were infected with RBSDV, respectively (Figure 3B, right, row 1). Then, the specificity of the PAb-CP^RSV^ and PAb-CP^RBSDV^ were validated by Western blot when diluted for 1 × 10^4^-fold. The Western blot results are in line with the RT-PCR, which showed that PAb-CP^RSV^ and PAb-CP^RBSDV^ can be used for RSV and RBSDV specific detection in SBPHs, and PAb-CP^RBSDV^ was not able to detect RSV, and vice versa (Figure 3B, right, row 2 and row 3). In summary, these results demonstrated that PAb-CP^RBSDV^ is highly specific and highly sensitive for the detection of RBSDV in rice plant and SBPHs, simultaneously.

### 3.4. Establishment of Specific ELISA Detection Methods

Furthermore, RBSDV- and RSV-infected rice seedlings were identified in the above Western blot as materials and the prepared specific recognition antibodies (PAb-CP^RBSDV^ and PAb-CP^RSV^) were added simultaneously, the ELISA detection method was developed for the specific detection of RBSDV or RSV in plant. In testing, the prepared PAb-CP^RBSDV^ was diluted 10^4^-fold, while the previously prepared PAb-CP^RSV^ [18] underwent a 10^5^-fold dilution. Color intensity was recorded at OD_405_. The results showed that when both PAb-CP^RBSDV^ and PAb-CP^RSV^ were used simultaneously, all the wells loaded with the total protein from RBSDV and RSV-positive rice plants exhibited an intense yellow color, which was consistent with the results of the respective positive controls (Figure 4A). The corresponding ELISA readings were collected, calculated, and statistically analyzed (Figure 4B,C).

The statistical data showed that when 1 × 10^5^-fold diluted PAb-CP^RSV^ and 1 × 10^4^-fold diluted PAb-CP^RBSDV^ were added at the same time, PAb-CP^RSV^ could specifically identify RSV-infected rice samples (1^#^, 2^#^ and 3^#^) and readings were similar to the positive samples (Figure 4B). At the same time, the readings of RBSDV-infected rice samples (4^#^, 5^#^ and 6^#^) were also similar to the positive control, but could not react with RSV-infected rice (Figure 4C), indicating that the prepared antibodies in the ELISA assay could only specifically identify RBSDV-infected rice, and were highly specific for detecting RBSDV-infected rice in the field.

### 3.5. Establishment of S-RT-LAMP Detection Method Based on Serological-Based and RT-LAMP

S-RT-LAMP experiments were performed with total RNAs extracted from immunoprecipitated products in potentially RBSDV-infected rice and SBPHs. After the RT-LAMP reaction with designed specific pairs of primers, the RBSDV-infected rice samples 1^#^, 2^#^, 3^#^, 4^#^ and SBPH samples 1^#^, 2^#^, 3^#^, 4^#^ exhibited ladder-like DNA fragments after agarose gel electrophoresis, identical to the positive sample (Figure 5A,C, row 1). Subsequently, after the addition of SYBR Green I nucleic acid dye to the RT-LAMP amplification products, the positive amplification products rapidly changed to green in color, while the orange color of the dye remained in the negative control, which, without amplification products (Figure 5B, row 1), indicated that the four rice seedlings were infected with RBSDV using the established S-RT-LAMP. Meanwhile, under UV light, the positive amplification products showed a strong fluorescence response, while no significant fluorescence was produced in the negative control (Figure 5B, row 2). S-RT-LAMP results of SBPHs samples carrying RBSDV are identical (Figure 5D). In sum, these results indicated that the established S-RT-LAMP method is suitable for a more sensitive and accurate detection of RBSDV in rice seedlings and SBPH in the wild field. 

## 4. Discussion

For viruses containing only one structural protein, a common way to prepare their antiserum is to apply their purified virions as an antigen in immune laboratory animals. However, it has been reported that purified RBSDV particles fail to elicit antibodies that specifically recognize the viral coat protein when injected to rabbits [22]. In addition, RBSDV is often restricted to the phloem of leaf veins in infected plants, which has also made it difficult to obtain mass quantities of highly purified virions, and also difficult to produce good quality antiserum for setting up efficient and economic detections [23]. However, the intact RBSDV virion has five structural proteins at least, among which the 63 kDa major outer capsid protein P10 is the most abundant [10]. As a result, we have explored the possibility of using this protein with His-tag to produce an antiserum for serological diagnosis.

In this study, the His-CP^RBSDV^ was purified (Figure 1C) and used as an antigen to immunize rabbits. Then, the crude antiserum against His-CP^RBSDV^ was then obtained (Figure 2). Then, the titer of crude antiserum was certified, and Western blot and dot-blot were used to test the quality of the antiserum using the purified His-CP^RBSDV^ as an antigen (Figure 2). After purification of the PAb-CP^RBSDV^ from the crude serum, we developed multiple serology-based methods for specific detection of RBSDV in two hosts with the polyclonal antibodies (Figure 3, Figure 4 and Figure 5). Despite the existence of various RBSDV antibodies that have been created and utilized for the identification of RBSDV in SBPHs and rice plant tissues [12,14,24], there has not been a comprehensive assessment of the precise potency of the developed antibodies against RBSDV in each detection method that has been employed. Therefore, the sensitivity and specificity of the PAb-CP^RBSDV^ were evaluated to determine the optimal titer, as well as specificity, for detecting RBSDV in the hosts.

In our study, we successfully developed Western blot, dot blot, ELISA, and RT-PCR detection methods utilizing the PAb-CP^RBSDV^ that we prepared. This polyclonal antibody-based serological approach can identify more epitopes than methods for preparing monoclonal antibodies for detection [12], allowing for a more efficient diagnosis of RBSDV in SBPH and rice seedling samples. It is worth noting that we identified the ideal titer of the prepared PAb-CP^RBSDV^ for rice seedling detection methods to be at a 10^4^-fold dilution (Figure 3). In addition, we used laboratory-prepared PAb-CP^RBSDV^ and PAb-CP^RSV^ for the simultaneous detection of rice and planthopper infected with either RSV or RBSDV, and the results also showed that the P10 antiserum prepared in this experiment was highly specific and significantly different from RSV for the detection of RBSDV in rice and SBPHs (Figure 3B). Hence, we used the ELISA method to specifically detect rice samples infected by RBSDV and RSV which can also sensitively detect these two viruses (Figure 4). The findings indicated a strong correlation between Western blot, dot-blot, RT-PCR, and ELISA techniques when used together, enabling a convenient and precise detection of RBSDV in both rice seedling samples and SBPHs found in natural environments.

Furthermore, we developed and optimized a method named S-RT-LAMP, which combines RT-LAMP with PAb-CP^RBSDV^ for the detection of RBSDV infected plants and insects. This study determined that total RNA isolated from products obtained by immunoprecipitation of infected RBSDV rice or SBPH with PAb-CP^RBSDV^ at 63 °C could be detected within only 60 min by RT-LAMP. During processes of total RNAs and protein extraction, CP^RBSDV^ might protect its genomic RNAs or viral transcripts via directly binding in cellular disruption condition. Hence, the S10 RNA here is more stable than that of direct total RNAs extraction. Furthermore, our method demonstrated remarkable sensitivity, surpassing that of the RT-PCR technique, as it successfully detected RBSDV in total RNA extracts up to 1.1 μg per reaction. This level of sensitivity highlights the superior capabilities of our approach in detecting the presence of RBSDV.

The key describable features of this S-RT-LAMP assay were: (1) we first used the prepared specific antibodies to immune-precipitate viral proteins (CP^RBSDV^) from plants and planthoppers, which ensures that the proteins packaged with viral RNA are directly isolated from the large amount of disrupted cellular components by immune-precipitation using PAb-CP^RBSDV^, which greatly avoided the degradation of the RNA S10 by RNase and improved the purity and quality of the samples for detecting viruses; (2) and then we combined RT-LAMP to achieve a large number of amplification efficiency under a short-term constant temperature conditions (Figure 5, row 1). Given that the reaction takes place under isothermal conditions at the ideal temperature, there is no time delay due to temperature fluctuations, similar to what is commonly observed in PCR [25]. (3) Additionally, there were many reports describing the use of RNA as an RT-LAMP template in one tube [16,26,27]. In our S-RT-LAMP, we employed crude purified RNAs that specifically bind to the CP^RBSDV^. This approach effectively mitigates the occurrence of false positives that may arise from contamination of the reaction product due to frequent lid opening. (4) Finally, with SYBR Green I used as the fluorescence indicator, the detection results could be directly judged by observing the color changes (Figure 5, row 2). In addition, fluorescence changes can also be observed under UV light conditions (Figure 5, row 3).

Based on the detection methods described above, the newly developed S-RT-LAMP was more suitable for direct and accurate detection of RBSDV on rice and SBPHs in the wild field in the present study. Our results of S-RT-LAMP possesses great potential in application for the field detection of RBSDV. Although there are certain complexities in the experimental process of S-RT-LAMP, due to its more stable experimental conditions and direct and efficient detection performance, the S-RT-LAMP assay could be a useful method for epidemiological research and for the forecast and control of the viruses’ diseases.

In addition, we only analyzed the specificity with RSV members of the genus *Fijivirus* that is in the same genus as RBSDV in this study, and we can also explore the cross-reaction between other members of the same genus, such as SRBSDV with a similar genomic structure to RBSDV [28] and high *CP* homology between the two species [29,30], according to which we speculate that cross-reactivity between them is highly likely. This requires follow-up experiments to prove it. In follow-up work, it is necessary to reflect the specificity of the prepared antibodies in additional diagnostics with different hosts to further analyze the differences that exist between different *Fijivirus* and to lay the groundwork for better detection and prediction of such *Fijivirus*.

## Figures and Tables

**Figure 1 viruses-15-02127-f001:**
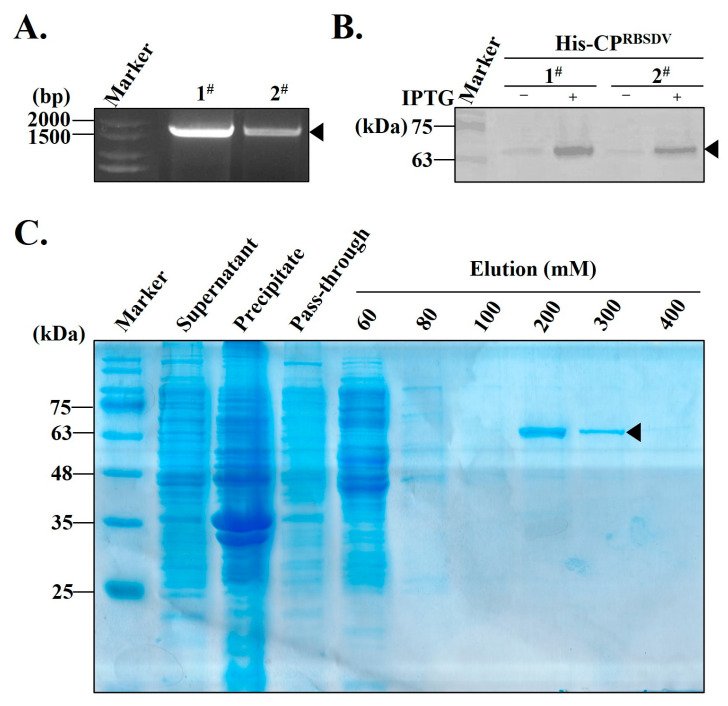
Construction of a prokaryotic expression vector pET28(a)-CP^RBSDV^ and purification of the recombinant His-CP^RBSDV^ protein. (**A**) Agarose-gel analysis of the amplification products by PCR from RBSDV-infected rice. Numbers 1^#^, 2^#^ represent 1677 bp *CP* coding region sequence in which its position is indicated by the black arrowhead. (**B**) Western blot analysis the expression of His-CP^RBSDV^ protein induced with (+) or without (−) IPTG. Numbers 1^#^, 2^#^ represent two-independent positive transformants by *E. coli* BL21 (DE3) transformants. (**C**) The prokaryotic expressed His-CP^RBSDV^ protein was purified using affinity chromatography. The bacterial suspension was subjected to ultrasonic breaking, followed by centrifugation to obtain the supernatant (lane 2), precipitate (lane 3), and pass-through (lane 4). Subsequently, the His-CP^RBSDV^ proteins were bound to the Ni-NTA resin through affinity binding. To elute the bound proteins, an imidazole solution with concentrations ranging from 60 mM to 400 mM was used (Lanes 5–10).

**Figure 2 viruses-15-02127-f002:**
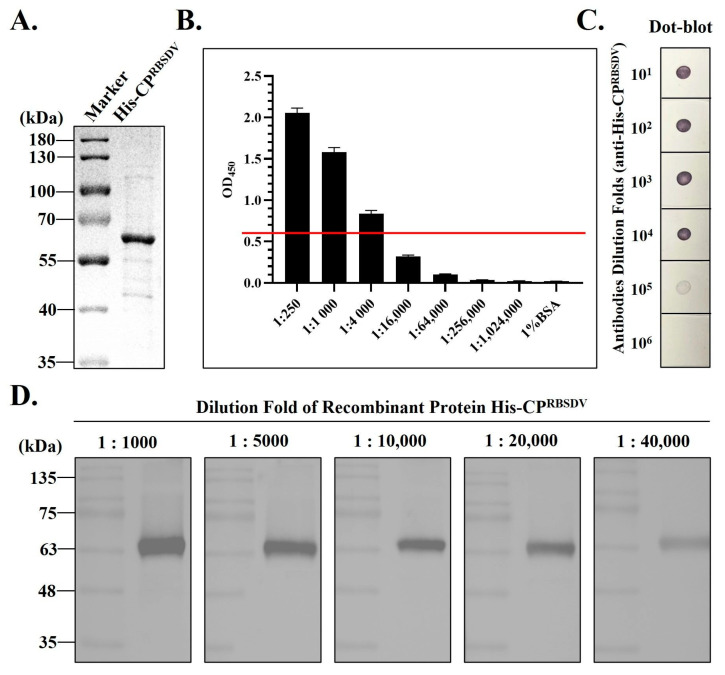
The prepared antiserum against the His-CP^RBSDV^ protein was evaluated, and three detection methods were used based on the His-CP^RBSDV^ protein. (**A**) SDS-PAGE analysis of the His-CP^RBSDV^ protein purified by affinity column. Marker stands for the protein marker. (**B**) Evaluation of the titer of CP^RBSDV^ specific antiserum was performed using ELISA. For the detection of the antigen-antibody reaction, HRP-A was used as the secondary antibody. The absorption value was measured under the wavelength of 450 nm. The red line position refers to the position where the OD_450_ is 0.6. (**C**) The antiserum was evaluated by PBS at different diluted concentrations in dot-blot assay. The purified His-CP^RBSDV^ was taken as antigen. (**D**) The Western blot was employed to detect the purified His-CP^RBSDV^ protein, using varying dilutions of PAb-CP^RBSDV^ antibody. Marker stands for the protein markers (Lane 1, 3, 5, 7, 9) and Lane 2, 4, 6, 8, 10 refer to the PAb-CP^RBSDV^ with 1 × 10^3^, 5 × 10^3^, 1 × 10^4^, 2 × 10^4^, and 4 × 10^4^ dilutions.

**Figure 3 viruses-15-02127-f003:**
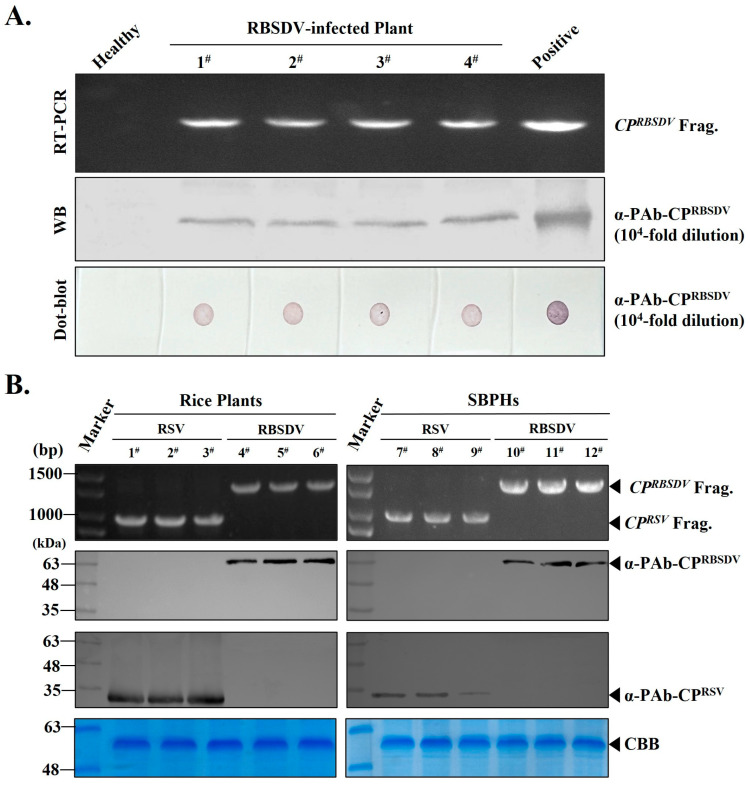
The purified PAb-CP^RBSDV^ could specifically detect the RBSDV with rice plants and SBPHs. (**A**) Multiple detections of the RBSDV-infected rice seedlings with prepared PAb-CP^RBSDV^. The uppermost panel showcases the outcome of the RT-PCR analysis, where lanes 2 to 4 pertain to four rice seedlings that were infected with RBSDV numbered 1^#^, 2^#^, 3^#^, 4^#^. Lane 1 corresponds to a healthy rice plant, while lane 5 serves as a reference for a purified His-CP^RBSDV^, which is utilized as a positive control. Following the RT-PCR results, the subsequent two panels present the outcomes of the Western blot and dot-blot assays, employing a diluted PAb-CP^RBSDV^ at 10^4^-fold dilution. (**B**) RT-PCR detection of the two rice viruses-infected rice plants and SBPHs with specific primers and Western blot detection of the two rice viruses-infected rice plants and SBPHs with both the prepared PAb-CP^RBSDV^ and PAb-CP^RSV^, respectively. First panel is the RT-PCR result, sample of 1^#^, 2^#^, 3^#^ refer to three RSV-infected rice seedlings, sample of 4^#^, 5^#^, 6^#^ refer to three RBSDV-infected rice seedlings, samples of 7^#^, 8^#^, 9^#^ refer to three RSV-infected SBPHs, and samples of 10^#^, 11^#^, 12^#^ refer to three RBSDV-infected SBPHs. The middle of these panels are the results of Western blot, samples are the same as the described above. The Rubisco large subunit, serving as a loading control, was subjected to Coomassie Brilliant Blue (CBB) staining for analysis.

**Figure 4 viruses-15-02127-f004:**
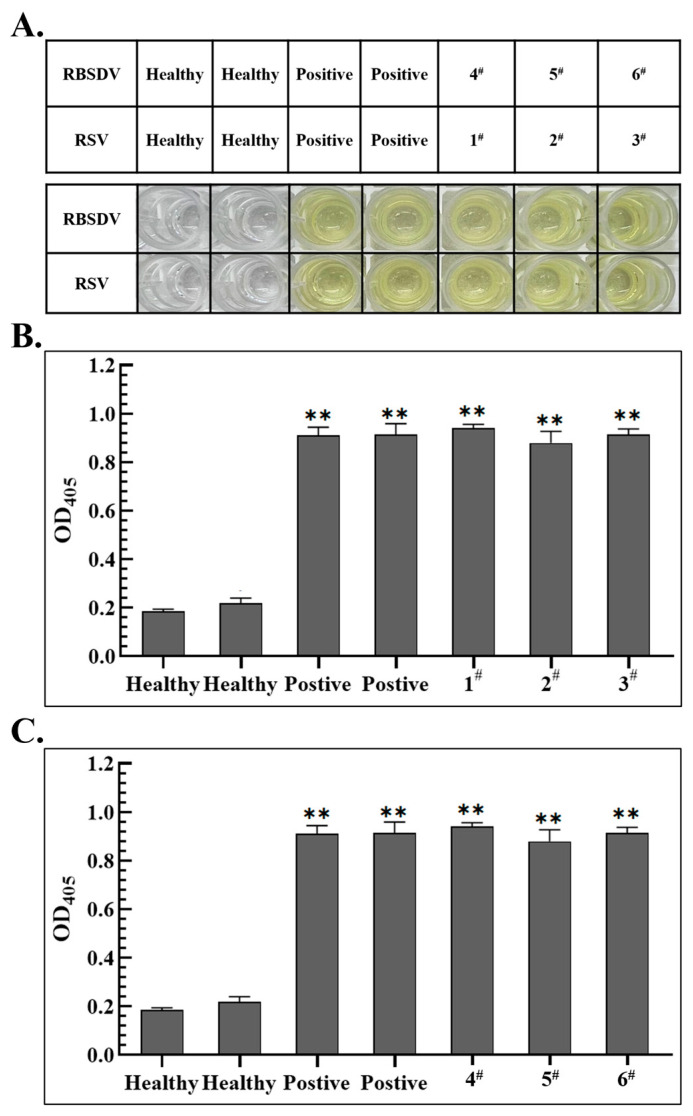
Application of the developed ELISA detection methods in the specific detection of RBSDV and RSV-infected rice seedlings in wild field. (**A**) The ELISA detection was employed to detect RBSDV and RSV infection in rice plants. The upper panel of the provided graphic displays the precise location of the samples that were loaded, while the bottom panel demonstrates the outcome of the blotting process. Lane 1 refers to the samples name, Lane 2–3 represents the negative control with healthy rice, Lane 4–5 refers to the positive control with RSV or RBSDV infected rice, Lane 6–8 refers to RSV-infected samples (1^#^, 2^#^, 3^#^) and RBSDV-infected samples (4^#^, 5^#^, 6^#^). (**B**,**C**) ELISA detection of the rice plants tested above with the PAb-CP^RSV^ and PAb-CP^RBSDV^. The absorbance values were read by the ELISA reader at OD_405_. *t*-test were performed to analyses the data, and two asterisks indicate significant difference with 95% confidence., “**” means significant difference with “*p* < 0.01”.

**Figure 5 viruses-15-02127-f005:**
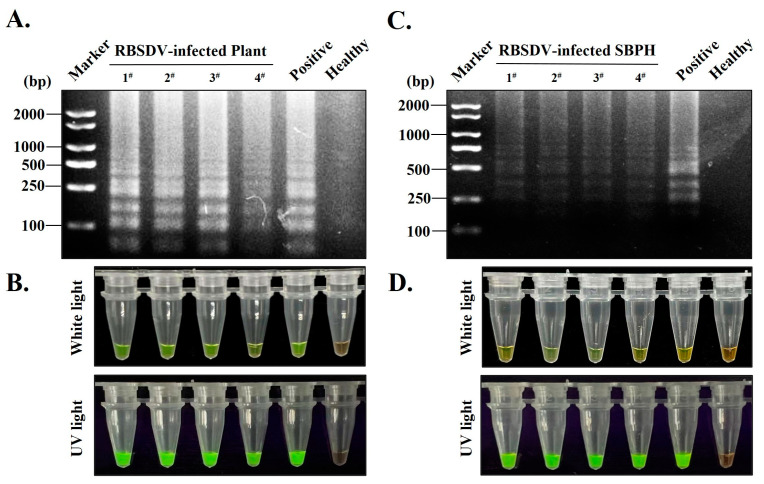
Application of the developed S-RT-LAMP detection methods in the sensitive detection of RBSDV-infected rice seedlings and SBPHs. (**A**,**C**) S-RT-LAMP reactions using total RNA of RBSDV-infected plant/planthopper isolated from precipitated by co-immunoprecipitation with the prepared PAb-CP^RBSDV^. First panel is the RT-LAMP results; Lane 2–5 refers to the samples infected by RBSDV of 1^#^, 2^#^, 3^#^ and 4^#^. Lane 6 and lane 7 refers to the positive control and negative control, respectively. Lane 1, DNA marker. (**B**,**D**) An addition of SYBR Green I into S-RT-LAMP reactions, and the color change and fluorescence degree at the bottom of the reaction tube, were directly observed in the white light. Turbid green fluorescence is a positive result, while orange-red is a negative result. The S-RT-LAMP products fluorescence degree at the bottom of the reaction tube were placed under the UV light. Positive results show strong green fluorescence, while negative results show no green fluorescence.

**Table 1 viruses-15-02127-t001:** List of the pairs of primer used in this study.

Primer Name	Primer Sequences (5′-3′) ^a^	Length of PCR Products (bp) ^b^	Tm Values (°C)	Purpose
RBSDV/P10/F	ATGGCTGACATAAGACTCG	1677	53.01	RT-PCR amplification of the S10 fragment of RBSDV
RBSDV/P10/R	CGCACAGCACTGAACTAGTC		57.45	
pET28a/P10/*EcoR* I/F	GGAATTCATGGCTGACATAAGACTCG	1677	59.59	Construction of the prokaryotic expression vector
pET28a/P10/*Xho* I/R	CCGCTCGAGTCTTGTCACTTTATTTAATAC		59.63	
RSV/NSvc3/F	ATGGGCACCAACAAGCCAGC	981	62.23	RT-PCR amplification of the NSvc3 fragment of RSV
RSV/NSvc3/R	GTCATCTGCACCTTCTGCCTC		59.14	

^a^. The letters that are underlined in the table indicate the specific sequences of the restriction enzyme site that have been inserted into the putative *CP* gene amplicon. ^b^. Genome position according to the reference nucleotide sequence of RBSDV, accession no. OR 395483.

**Table 2 viruses-15-02127-t002:** The four oligonucleotide primers used for the S-RT-LAMP assay for the detection of RBSDV.

Primer Name	Type Primer	Length of Primers	Genome Position
F3	Forward outer	21-mer	279–299
B3	Backward outer	21-mer	467–447
FIP (F1c + F2)	Forward inner	45-mer	366–344, 304–325
BIP (B1c + B2)	Backward inner	47-mer	374–398, 444–423

## Data Availability

Appendix A was listed in Appendix A.

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
