# Peer review of "Development of Polyclonal Antibodies and a Serological-Based Reverse-Transcription Loop-Mediated Isothermal Amplification (S-RT-LAMP) Assay for Rice Black-Streaked Dwarf Virus Detection in Both Rice and Small Brown Planthopper"

_viruses, 2023, doi:10.3390/v15102127_

Round 1

Reviewer 1 Report

Authors described in the manuscript the method to obtain valuable antibodies to diagnose RBSDV by Western blot dot blot and Elisa. The method described in the manuscript is the same as the one used for RSV diagnosis, see reference below, only the S-RT-Lamp is new. So the author needs to refer to Zhang et al in material and method for all step of antibody preparation and immunology diagnosis. And briefly describe the method with the modification made to what was described before. Otherwise, authors showed specificity of their antibody using RSV as test. This specificity is clearly shown for Western blot but not for Elisa where none cross reaction was tested or illustrated. Also there are errors in figure 4 and 5.  All comments are already in attached file with other comments and editing proposition.

Zhang, K., Zhuang, X., Xu, H., Guo, X., He, Z., Xu, K., & Liu, F. (2021). Sensitive and high-throughput polyclonal antibody-based serological methods for rice stripe virus detection in both rice and small brown planthopper. Crop Protection, 144, 105599.

english quality needs to be enhanced

Author Response

Response to Reviewer #1

Dear Editors and Reviewers:

Thank you for your letter and for the reviewers’ comments concerning our manuscript entitled “Development of polyclonal antibodies and serological-based reverse-transcription loop-mediated isothermal amplification (S-RT-LAMP) assay for rice black-streaked dwarf virus detection in both rice and small brown planthopper” (ID: viruses-2647735). These comments are all valuable and very helpful for revising and improving our paper, as well as the important guiding significance to our researches. We have studied comments carefully and have made correction which we hope meet with approval. The main corrections in the paper and the responds to the reviewer’s comments are as following:

  1. Authors described in the manuscript the method to obtain valuable antibodies to diagnose RBSDV by Western blot dot blot and Elisa. The method described in the manuscript is the same as the one used for RSV diagnosis, see reference below, only the S-RT-Lamp is new. So the author needs to refer to Zhang et al in material and method for all step of antibody preparation and immunology diagnosis. And briefly describe the method with the modification made to what was described before.

Response:

We sincerely appreciate the valuable comments. Firstly, we have checked the literature carefully and added more relevant references as followed into the first paragraph of Materials and Methods part in the revised manuscript. Secondly, the differences between the relevant experimental operations and those previously reported in the literature are also briefly described in the Materials and Methods section of the revised manuscript.

Relevant References:

[18] Kun, Z.; Hongmei, X.; Xinjian, Z.; Xiao, G.; Zhen, H.; Kai, X.; Fang, L., Sensitive and high-throughput polyclonal antibody-based serological methods for rice stripe virus detection in both rice and brown planthopper. Crop Protection 2021.

[19] Zhang, K.; Zhuang, X.; Xu, H.; Gan, H.; He, Z.; Chen, J., Development of polyclonal antibodies-based serological methods and a DIG-labelled DNA probe-based molecular method for detection of the Vicia cryptic virus-M in field plants. J Virol Methods 2022, 299, 114331.
[20] Jiahuan, C.; Chenwei, F.; Xiao, G.; Yuchen, Z.; Tianxiao, G.; Xinjian, Z.; Lei, C.; Kun, Z., Development of polyclonal antibodies-based serological methods for detection of the rehmannia mosaic virus in field plants&#13. Frontiers in Sustainable Food Systems 2022, 6.

  1. Otherwise, authors showed specificity of their antibody using RSV as test. This specificity is clearly shown for Western blot but not for Elisa where none cross reaction was tested or illustrated. Also there are errors in figure 4 and 5.

Response:

Thank you for your valuable comments on the " 3.4 Establishment of specific ELISA detection methods " section of the experiment, and sorry for the misunderstanding caused by the description in this section. We apologize that the samples used in this experiment are from samples collected in Western Blot above, and the antibodies added are not clearly stated. Hence, we rephrase this conclusion in the Results section, and we intend to use known positive sample material to specifically detect the prepared antibodies. In this way, it is shown that the antibody we prepared can also specifically recognize RBSDV in rice in ELISA experiments. Please see page 9 of the revised manuscript, lines 346-364. Some of the errors in Figures 4 and 5 have also been corrected.

  1. All comments are already in attached file with other comments and editing proposition.

Response:

Thank you for your valuable revisions to the details in the manuscript. We have revised the other changes mentioned in the attached file as required. And we explain the sections where the specific content has been modified and adjusted as followed.

In the Introduction section, we apologize for the ambiguity caused by the statement that "there is no method to detect rice and planthopper infected with RBSDV at the same time". At the same time, we have also added relevant literature to illustrate this issue in accordance with the recommendations. Please see page 2 of the revised manuscript, lines 75-81.

In the Materials and Methods section, we add relevant references before each experimental method. In addition, the acquisition of PAb-CPRSV used in subsequent experiments is also explained in lines 132-134 on page 3 of the revised manuscript.

In the Results section, we have carefully revised every legend and description of errors.

In the Discussion section, we have supplemented the parts that lack the support of the literature in line 422-424 on page 12 of the revised manuscript. And then there is one point where the expression is not clear. We replaced “Maybe the protection of the CPRBSDV protein in cellular disruption condition, and the S10 RNA is stable than that of directly total RNAs extraction.” with “During processes of total RNAs and protein extraction, CPRBSDV maybe protect its genomic RNAs or viral transcripts via directly binding in cellular disruption condition. Hence, the S10 RNA here is more stable than that of from directly total RNAs extraction.” by highlight in lines 448-451 on page 12 of the revised manuscript.

Thank you very much for your attention and time. Look forward to hearing from you.

Yours sincerely,

Yanhong Hua

11 Oct 2023

Yangzhou University

Reviewer 2 Report

This study was well conducted overall and the conclusions were supported by the results presented. The only major comments are as follows.

Line 405, 488

S-RT-LAMP, which is based on immune-captured nucleic acid amplification, is an excellent method without nonspecific reactions. However, compared to conventional RT-LAMP methods, which use crude sap extract as a template, the experimental process is rather complicate, and it does not seem to be the rapid and simple assay as claimed by the authors. It would be better to improve the description for the convenience of this method in discussion a little more.

Although the authors maintain the specificity of the antibodies produced, it is also necessary to investigate the specificity against other Fijivirus viruses. In particular, SRBSDV is serious viral disease of rice in East Asia, although the vector insect is different. The homology of CP is approximately 80% between RBSDV and SRBSDV. Therefore, the antibody produced in this study may cross-react with SRBSDV CP antigen. Even if cross-reaction happened, it would not decrease the value of this paper. I recommend that the authors describe the necessity to investigate the possibility of cross-reaction with other Fijiviruses in the future in the discussion.

Table S1 and S2

I will recommend that the primers designed in this study to detect this virus for RT-PCR and RT-LAMP would be described in the main text rather than in the supplement tables. Please consider it.

Line 153

Infest: wrong, infect: correct

Line 335

It would be better to briefly describe how PAb-CP-RSV was made. Is the antibody also produced using a recombinant protein derived from the expression vector?

That is all.

Author Response

Response to Reviewer #2

Dear Editors and Reviewers:

Thank you for your letter and for the reviewers’ comments concerning our manuscript entitled “Development of polyclonal antibodies and serological-based reverse-transcription loop-mediated isothermal amplification (S-RT-LAMP) assay for rice black-streaked dwarf virus detection in both rice and small brown planthopper” (ID: viruses-2647735). These comments are all valuable and very helpful for revising and improving our paper, as well as the important guiding significance to our researches. We have studied comments carefully and have made correction which we hope meet with approval. The main corrections in the paper and the responds to the reviewer’s comments are as following:

  1. Line 405, 488, S-RT-LAMP, which is based on immune-captured nucleic acid amplification, is an excellent method without nonspecific reactions. However, compared to conventional RT-LAMP methods, which use crude sap extract as a template, the experimental process is rather complicate, and it does not seem to be the rapid and simple assay as claimed by the authors. It would be better to improve the description for the convenience of this method in discussion a little more.

Response:

Thank you for your valuable suggestions on the presentation of the issues in this part of the experiment and discussion. As you said, compared to conventional RT-LAMP with crude juice extract as a template, S-RT-LAMP ensures the reliability of RNA sources by combining serological methods with RT-LAMP, so the accuracy of detection will be greatly improved. Hence, this may also increase the complexity of the work correspondingly. So we modified the description of this method. Please see page 11 (lines 389-391) and page 13 (lines 475-478) of the revised manuscript.

  1. Although the authors maintain the specificity of the antibodies produced, it is also necessary to investigate the specificity against other Fijivirus viruses. In particular, SRBSDV is serious viral disease of rice in East Asia, although the vector insect is different. The homology of CP is approximately 80% between RBSDV and SRBSDV. Therefore, the antibody produced in this study may cross-react with SRBSDV CP antigen. Even if cross-reaction happened, it would not decrease the value of this paper. I recommend that the authors describe the necessity to investigate the possibility of cross-reaction with other Fijiviruses in the future in the discussion.

Response:

Thank you for your valuable comments. Since RBSDV and SRBSDV have similar genomic structures, and their CPsequences also have extremely high homology, as stated in the relevant literature [28-30]. Wang et al. (2010) have reported that ‘The isolates were most closely related to Rice black-streaked dwarf virus (RBSDV): S1, S2 and S10 were most conserved with identities of 78.5–79.2% nt (83.4–89.0% aa) while S5 and S6 were the least conserved with 70.6–71.6% nt (63.1–69.9% aa) identity’. Therefore, we speculate that it is also possible for the prepared PAb-CPRBSDV antibody to react with the SRBDV sample. So we have added to our discussions an outlook on what can be explored in the future. Please see page 13 (lines 479-487) of the revised manuscript.

Relevant References:

[28] Zhang, P.; Mar, T. T.; Liu, W.; Li, L.; Wang, X., Simultaneous detection and differentiation of Rice black streaked dwarf virus (RBSDV) and Southern rice black streaked dwarf virus (SRBSDV) by duplex real time RT-PCR. Virol J 2013, 10, 24.

[29] Cheng, Z.; Li, S.; Gao, R.; Sun, F.; Liu, W.; Zhou, G.; Wu, J.; Zhou, X.; Zhou, Y., Distribution and genetic diversity of Southern rice black-streaked dwarf virus in China. Virol J 2013, 10, 307.

[30] Wang, Q.; Yang, J.; Zhou, G.; Zhang, H.; Chen, J.; Adams, M. J., The Complete Genome Sequence of Two Isolates of Southern rice black-streaked dwarf virus, a New Member of the Genus Fijivirus. Journal of Phytopathology 2010, 158, 733-737.

  1. Table S1 and S2, i will recommend that the primers designed in this study to detect this virus for RT-PCR and RT-LAMP would be described in the main text rather than in the supplement tables. Please consider it.

Response:

Thank you for your valuable comments. We have made modifications based on your comments. Please see page 3 (line 113) and 5 (line 223) of the revised manuscript. And the corresponding legend was changed in the revised manuscript.

  1. Line 153, Infest: wrong, infect: correct

Response:

We feel sorry for our carelessness. In our resubmitted manuscript, the typo is revised. Thanks for your correction.

  1. Line 335, It would be better to briefly describe how PAb-CP-RSV was made. Is the antibody also produced using a recombinant protein derived from the expression vector?

Response:

Thank you for your valuable question. The method of obtaining PAb-CPRSV is similar to the method of obtaining polyclonal antibodies against recombinant His-CPRBSDV protein, both of which obtain recombinant His-CPRSV or His-CPRBSDV protein by constructing prokaryotic expression vectors, and then obtaining corresponding antibodies by immunizing New Zealand white rabbits. We add a specific description to “ 2.3. Preparation of the polyclonal antibody against the recombinant His-CPRBSDV protein” of Methods and Materials in lines 132-134 on page 3 of the revised manuscript.

Thank you very much for your attention and time. Look forward to hearing from you.

Yours sincerely,

Yanhong Hua

11 Oct 2023

Yangzhou University

Round 2

Reviewer 1 Report

Dear Authors, the manuscript was accuratly modified but there are still some editing modification to be made. Also author names in reference18  were miss written.

Needs some corrections

Author Response

Response to Reviewer #1

Dear Editors and Reviewers:

Thank you for your letter and for the reviewers’ comments concerning our manuscript entitled “Development of polyclonal antibodies and serological-based reverse-transcription loop-mediated isothermal amplification (S-RT-LAMP) assay for rice black-streaked dwarf virus detection in both rice and small brown planthopper” (ID: viruses-2647735). These comments are all valuable and very helpful for revising and improving our paper, as well as the important guiding significance to our researches. We have studied comments carefully and have made correction which we hope meet with approval. The main corrections in the paper and the responds to the reviewer’s comments are as following:

  1. Dear Authors, the manuscript was accurately modified but there are still some editing modification to be made. Also author names in reference 18 were miss written.

Response: Thank you again for your reply and careful review. We apologize for the miss of the author's name from reference 18, and recheck and revise the cited document in the revised manuscript. Please see lines 542-544 on page 14 of the revised manuscript. Other citied references were also have been checked from aspects of spelling, formats, and the detailed information.

In addition, we also check and modify according to the specific changes proposed in the attached file. Your careful revision of this manuscript greatly improves the accuracy of the description, and the specific changes are as follows.

In the Introduction section, we adjusted the section describing the symptoms of RBSDV based on your recommendations. At the same time, we have also modified the description of the viral structure of RBSDV. Please see lines 48-50 and lines 61-62 on page 2 of the revised manuscript, respectively. Additionally, we adjusted the description of the detected object and methods to avoid ambiguity. Please see line 77, line 87 and lines 93-94 on page 2 of the revised manuscript.

In the Materials and Methods section, we feel sorry for our carelessness. In our resubmitted manuscript, the typo is revised. Thanks for your correction. Please see line 100 on page 3 of the revised manuscript.

In the Results section, we apologize for using the phrase incorrectly. It has been revised in the revised manuscript in line 354 on page.

In the Discussion section, we provide an accurate description of the method of reporting as recommended. Please see line 462 on page 13 of the revised manuscript. Additionally, we're sorry for the incorrect use of prepositions. We have corrected them in line 472 and 476 on page 13. We have supplemented the specific descriptions referred to two species. The term for other methods was replaced with a more specific diagnostic method. Please see line 479 and 482 on page 13 of the revised manuscript.

Thank you very much for your attention and time again. Look forward to hearing from you.

Yours sincerely,

Kun Zhang

16 Oct 2023

Yangzhou University
